# Impact of the use of GeneXpert on TB diagnosis and anti-TB treatment outcome at health facilities in Addis Ababa, Ethiopia in the post-millennium development years

**Desalegn Addise Getahun**[1,2]*, **Laura E. Layland**[2], **Achim Hoerauf**[2,3], **Biniam Wondale**[4]

**1** Ethiopian Public Health Institute, Addis Ababa, Ethiopia, **2** Institute of Medical Microbiology, Immunology and Parasitology, University Hospital Bonn, Bonn, Germany, **3** German Center for Infection Research (DZIF), Partner Site Bonn-Cologne, Bonn, Germany, **4** Department of Biology, Arba Minch University, Arba Minch, Ethiopia

* desalegnaddise@gmail.com

## Abstract

### Background

GeneXpert is an effective and rapid molecular system used for tuberculosis (TB) diagnosis. It is expected to improve the detection rate and treatment outcomes needed to meet the sustainable development goals (SDG) and End TB strategy targets set for 2030. This study aimed to evaluate the impact of GeneXpert on diagnosis and anti-TB treatment outcomes in the post-millennium development goals (MDGs) in the capital city of Ethiopia. Hence, the global priority indicator based on the End TB Strategy for TB treatment success rate was met early in 2018 in Addis Ababa, Ethiopia, which was anticipated to be met by 2025.

### Methods

A retrospective health facilities-based study was conducted in Addis Ababa, Ethiopia. Records of all TB cases diagnosed and treated in selected health facilities from January 1st, 2015 to December 31st, 2018 were reviewed and included in the study. Data analysis of descriptive and inferential statistics was conducted using SPSS version 20.

### Results

The reviewed records have shown that a total of 45,158 presumptive pulmonary TB (PTB) cases had accessed TB diagnosis services. Of which, 28.9% (13072/45158) were tested by AFB microscopy and 71.1% (32086/45158) were tested by GeneXpert. During the study period, the coverage of Xpert MTB/RIF testing increased to 94.9% in 2018 compared to 1.6% in 2015. The number of presumptive PTB cases tested with the GeneXpert system showed a significant increase compared to smear microscopy. The odds of positivity were detected in males compared to females. The odds of detecting TB cases were much higher among study participants aged 15–44 years compared to younger than 15 years. Treatment success rate showed a relative improvement each year between 2015 and 2018 with a

**Data Availability Statement:** All relevant data are within the paper and its Supporting information files.

**Funding:** The first author, Desalegn Addise, was funded by the Uni Bonn for the Master's program that helped enable him to conduct this study and Achim Hoerauf is also funded by the Deutsche Forschungsgemeinschaft (DFG, German Research Foundation) under Germany's Excellence Strategy – EXC2151 – 390873048. The funders had no role in study design, data collection and analysis, decision to publish, or preparation of the manuscript.

**Competing interests:** The authors have declared that no competing interests exist.

mean of 92.6%. Reduced odds of treatment successes were observed in age categories older than 35 years, and in TB/HIV co-infected patients. Increased odds of treatment successes were reported in the years between 2016 and 2018 compared to 2015.

## Conclusion

Scaling up the Xpert MTB/RIF assay as a point-of-care test for presumptive TB cases in resource-limited settings would have a significant impact to meet the SDG and End TB strategy both in TB detection and treatment success rates.

## 1. Background

Tuberculosis (TB) is a chronic infectious disease mainly caused by *Mycobacterium tuberculosis* that affects primarily the lung. There were an estimated 10 million TB cases in 2019. There were 206,030 multi-drug resistant (MDR) cases including rifampicin resistant (RR), and 12,350 laboratory-confirmed extensively drug resistant (XDR) TB cases. In the same year, about 1.4 million people died from TB [1].

In sub-Saharan African countries, the rapid increase in drug-resistant TB cases is a challenge [2]. The Xpert MTB/RIF assay is a solution to detect *M. tuberculosis* including RR cases at the same time [3, 4]. In sputum samples with negative smear results, the Xpert MTB/RIF assay has a pooled sensitivity of 60.6–77% and specificity of 98.8% [5]. Due to its improved sensitivity and specificity over smear microscopy for the diagnosis of TB, the Xpert MTB/RIF assay improves the anti-TB treatment outcome [6–9]. The Xpert MTB/RIF assay reduces the anti-TB treatment initiation time by around four weeks for patients with paucibacillary for *M. tuberculosis* [6, 10]. Given this advance, the World Health Organization (WHO) endorsed and recommended the use of the GeneXpert system at the point of care for the diagnosis of TB [11].

Early detection and treatment of TB are crucial for the effort towards meeting the sustainable development goal (SDG) and End TB strategy targets set for 2035. Ethiopia achieved the TB-related Millennium Development Goals (MDGs) with a reduction of deaths due to TB by 75% and the TB incidence rate by more than 50% in 2015 compared to 1990 [2]. The GeneXpert instruments were introduced and have been in use in Ethiopia since 2012 [12]. They are used to diagnose only selected presumptive TB cases (presumptive DR-TB, HIV positive, children, and extrapulmonary TB), while the remaining cases with low-risk for DR-TB, HIV negative, and adult are diagnosed by sputum-smear microscopy [13]. Sputum-smear microscopy is a first-line test despite having a very poor sensitivity (25–75% compared to culture) and being unable to detect drug-resistant strains [14]. Since 2017, the GeneXpert system has been in use to diagnose all presumptive TB cases presenting with signs and symptoms of TB at any given health facility in Addis Ababa. However, the impact of Xpert MTB/RIF assay in TB diagnosis and treatment outcomes in health facilities of Addis Ababa has not been evaluated. Therefore, this study aimed to review the available records at the selected health facilities in Addis Ababa, Ethiopia, and to evaluate the impacts of Xpert MTB/RIF assay on the number of TB cases detected and successes in anti-TB treatment outcomes.

## 2. Methods

### 2.1 Study setting

The study was conducted in 19 public health facilities, which provide TB diagnosis services using the GeneXpert system in Addis Ababa, Ethiopia. Addis Ababa is the capital city of

Ethiopia, covering an area of 540 km$^2$ with an estimated population of 3,384,569 according to the 2007 census (World population review 2020). The city is divided into 10 administrative sub-cities.

The government health infrastructure of Addis Ababa includes hospitals, health centers, and Addis Ababa Public Health Research and Emergency Management (AAPHREM).

## 2.2 Study design, data collection, and ethical review

A retrospective health facility-based study was conducted in the capital city of Ethiopia by reviewing the laboratory TB registry logbook and TB unit register. In Addis Ababa, the Xpert MTB/RIF testing service is provided in five hospitals, 18 health centers, and one regional reference laboratory. The type of methods used for TB diagnosis and the number of presumptive pulmonary TB cases tested in the study period between January 1$^{st}$, 2015 and December 31$^{st}$, 2018 were recorded from the laboratory TB registry logbook. The data for this study were retrieved between December 1, 2019 and February 28, 2020. All tests for all age groups of presumptive pulmonary TB cases were done using sputa as a sample source. All TB cases on anti-TB treatment, which were registered in the respective health facilities' TB unit register during the study period, were included in the study. Out of 24 health facilities, which provided Xpert MTB/RIF testing services, five health facilities were excluded from the study. Of the five excluded health facilities, four health facilities did not document TB registry logbooks and one health facility refused to participate in the study. An additional two health facilities were excluded from treatment outcome analysis since they only provided test service but not direct observed treatment short-course (DOT) service. Hence, the laboratory data were collected from 19 health facilities whereas anti-TB treatment outcome data were collected from 17 health facilities.

Demographic and clinical data of TB cases were reviewed and recorded into Microsoft Excel by a data clerk. The correctness of the recorded data was cross-checked by the principal investigator.

Ethical clearance was obtained from the Ethical Review Committee of the University Hospital Bonn and the AAPHREM. In addition, a support letter was obtained from the University Hospital Bonn, Institute of Hygiene and Public Health. The selected health facilities were informed about the study and written consent was obtained. The collected data privacy and confidentiality were protected at all times during the data collection period. All participants' identifiers were removed during analysis once the data collection and quality assurance processes were completed.

**2.2.1 Operational definitions.** The following treatment outcomes were defined as follows per WHO guideline 2013 [15].

Cured–A pulmonary TB patient with bacteriologically confirmed TB at the beginning of treatment who was smear- or culture-negative in the last month of treatment and on at least one previous occasion.

Treatment completed–A TB patient who completed treatment without evidence of failure BUT with no record to show that sputum smear or culture results in the last month of treatment and on at least one previous occasion were negative, either because tests were not done or because results are unavailable.

Treatment failed–A TB patient whose sputum smear or culture is positive at month 5 or later during treatment.

Died–A TB patient who dies for any reason before starting or during the course of treatment.

Loss to follow-up–A TB patient who did not start treatment or whose treatment was interrupted for two consecutive months or more.

Not evaluated—A TB patient for whom no treatment outcome is assigned. This includes cases "transferred out" to another treatment unit as well as cases for whom the treatment outcome is unknown to the reporting unit. Transfer-in—patients have been transferred from another TB register to continue treatment.

Successful treatment outcome–the sum of cured and treatment completed.

### 2.3 Data analysis and statistics

The data were primarily recorded in Microsoft Excel and then exported into SPSS Version 20.0 software (SPSS INC, Chicago, IL, USA) for analysis. The findings were presented using descriptive statistics. Bivariate and multivariate logistic regression models were also used to evaluate the association of independent variables with dependent variables. The level of significance was set at $P \leq 0.05$.

## 3. Results

### 3.1 Trends of presumptive pulmonary TB case diagnosis from smear microscopy to GeneXpert system

The reviewed records show that a total of 45,158 presumptive pulmonary TB cases had accessed TB diagnosis services. Of the total presumptive pulmonary TB cases in the study period, 28.9% (13072/45158) were tested by acid-fast bacilli (AFB) microscopy whereas 71.1% (32086/45158) were tested using the GeneXpert system Table 1. Xpert MTB/RIF testing coverage increased during the study period from 1.6% in 2015 to 94.9% in 2018.

### 3.2 TB positivity and associated factors within presumptive TB cases

After adjusting for factors associated with TB case detection, the GeneXpert system had higher odds of detecting TB cases (AOR: 4.78, 95% CI: 4.13–5.53) compared to smear microscopy. Presumptive TB cases were more likely to be diagnosed with TB if they were male (AOR: 1.45, 95% CI: 1.36–1.54) compared to female cases. The odds of detecting TB cases increased among study participants aged 15–44 years compared to study participants younger than 15 years Table 2.

### 3.3 Demographics and clinical characteristics of study participants on anti-TB treatment

The total numbers of patients who received anti-TB treatment in 17 health facilities were 8446 including extrapulmonary TB (EPTB) cases. Table 3 details the demographic and clinical characteristics of the survey TB patients who received anti-TB treatment during the study periods.

**Table 1. Uptake of GeneXpert system compared to smear microscopy for TB diagnosis at Addis Ababa, Ethiopia, between 2015 and 2018.**

| Year | Method of Diagnosis | | Total N (%) |
|---|---|---|---|
| | Microscopy N (%) | GeneXpert system N (%) | |
| 2015 | 4560 (98.4) | 75 (1.6) | 4635 (10.3) |
| 2016 | 5679 (61.2) | 3597 (38.8) | 9276 (20.5) |
| 2017 | 1931 (14.1) | 11744 (85.9) | 13675 (30.3) |
| 2018 | 902 (5.1) | 16670 (94.9) | 17572 (38.9) |
| Total | 13072 (28.9) | 32086 (71.1) | 45158 (100.0) |

**Table 2. Factors associated with the positive results of presumptive TB cases in Addis Ababa, Ethiopia between 2015 and 2018.**

| Variable | Total N (%) | Positive case N (%) | COR (95% CI) | AOR (95% CI) | P-value |
|---|---|---|---|---|---|
| Sex[a] | | | | | |
| F | 22882 (50.7) | 2081 (9.1) | 1 | 1 | |
| M | 22242 (49.3) | 2694 (12.1) | **1.38 (1.30–1.46)** | **1.45 (1.36–1.54)** | <0.001 |
| Age[b] | | | | | |
| 0–14 | 2309 (5.1) | 113 (4.9) | 1 | 1 | |
| 15–24 | 8359 (18.6) | 1273 (15.2) | **3.49 (2.86–4.26)** | **3.72 (3.05–4.55)** | <0.001 |
| 25–34 | 11769 (26.2) | 1582 (13.4) | **3.02 (2.48–3.67)** | **3.03 (2.48–3.69)** | <0.001 |
| 35–44 | 8075 (18.0) | 822 (10.2) | **2.20 (1.80–2.70)** | **2.08 (1.69–2.55)** | <0.001 |
| > = 45 | 14436 (32.1) | 809 (5.6) | 1.15 (0.94–1.41) | 1.11 (0.91–1.36) | 0.313 |
| Methods of Diagnosis | | | | | |
| Microscopy | 13072 (28.9) | 846 (6.5) | 1 | 1 | |
| GeneXpert system | 32086 (71.1) | 3963 (12.4) | **2.04 (1.89–2.20)** | **4.78 (4.13–5.53)** | <0.001 |
| Year of Diagnosis | | | | | |
| 2015 | 4635 (10.3) | 455 (9.8) | 1 | 1 | |
| 2016 | 9276 (20.5) | 1167 (12.6) | **1.32 (1.18–1.48)** | **0.72 (0.62–0.83)** | <0.001 |
| 2017 | 13675 (30.3) | 1525 (11.2) | **1.15 (1.03–1.29)** | **0.37 (0.31–0.44)** | <0.001 |
| 2018 | 17572 (38.9) | 1662 (9.5) | 0.96 (0.86–1.07) | **0.28 (0.24–0.34)** | <0.001 |
| Sub-cities | | | | | |
| Addis Ketema | 3564 (7.9) | 284 (8.0) | 1 | 1 | |
| Kolfe Keranyo | 8101 (17.9) | 744 (9.2) | **1.17 (1.01–1.35)** | **1.53 (1.31–1.77)** | <0.001 |
| Bole | 1645 (3.6) | 194 (11.8) | **1.54 (1.27–1.87)** | **1.43 (1.17–1.75)** | <0.001 |
| Akaki-Kality | 4759 (10.5) | 524 (11.0) | **1.43 (1.23–1.66)** | **1.59 (1.36–1.86)** | <0.001 |
| Yeka | 7980 (17.7) | 683 (8.6) | 1.08 (0.94–1.25) | **1.23 (1.06–1.44)** | 0.006 |
| Arada | 4897 (10.8) | 470 (9.6) | **1.23 (1.05–1.43)** | **1.41 (1.20–1.65)** | <0.001 |
| Nifas Silk | 7028 (15.6) | 710 (10.1) | **1.30 (1.12–1.50)** | **1.48 (1.28–1.72)** | <0.001 |
| Kirkos | 1704 (3.8) | 147 (8.6) | 1.09 (0.89–1.34) | **1.32 (1.06–1.64)** | 0.013 |
| AAPHREM* | 5480 (12.1) | 1053 (19.2) | **2.75 (2.39–3.16)** | **1.42 (1.20–1.69)** | <0.001 |

[a]- of the total 34 participants had no sex information;

[b]- of the total 210 participants had no age information;

* stands for Addis Ababa Public Health Research and Emergency Management.

The majority, 77% (6504/8446) of TB patients in the survey were in the age group 15–44 years. Male patients 51.7% (4365/8446), who received treatment in the study period were slightly higher than females. The percentages of patients with no anti-TB treatment history, re-treatment, and transfer-in cases were 86.3% (7282/8446), 9.4% (797/8446), and 4.3% (367/8446), respectively. TB/HIV co-infection was 20.4% (1723/8446). TB cases in the study period were Pulmonary TB bacteriologically negative (PTBN) 26.7% (2252/8446), Pulmonary TB bacteriologically positive (PTBP) 29.6% (2500/8446), and EPTB 43.7% (3694/8446).

### 3.4 Treatment outcomes

Overall, 7478 out of 8080 TB patients (92.6%) had successful treatment outcomes. Of these, 27.3% (2205/8080) patients had been cured and 65.3% (5273/8080) patients had completed treatment. HIV-infected TB patients had a lower rate of treatment success 86.8% (1416/1632) compared to non-infected ones 94.2% (5765/6121). Treatment outcomes after 2015 showed relative improvement Table 4.

**Table 3. Demographics and clinical characteristics of the survey TB patients who received anti-TB treatment in Addis Ababa, Ethiopia between 2015 and 2018.**

| Variables | TB Types in Number (%) | | | Total N (%) |
|---|---|---|---|---|
| | PTBN | PTBP | EPTB | |
| Sex | | | | |
| F | 992 (44.0) | 1097 (43.9) | 1992 (53.9) | 4081 (48.3) |
| M | 1260 (56.0) | 1403 (56.1) | 1702 (46.1) | 4365 (51.7) |
| Age | | | | |
| < 15 | 84 (3.7) | 53 (2.1) | 178 (4.8) | 315 (3.7) |
| 15–24 | 533 (23.7) | 763 (30.5) | 1012 (27.4) | 2308 (27.3) |
| 25–34 | 596 (26.5) | 895 (35.8) | 1213 (32.9) | 2704 (32.0) |
| 35–44 | 431 (19.1) | 418 (16.7) | 643 (17.4) | 1492 (17.7) |
| 45–54 | 257 (11.4) | 202 (8.1) | 283 (7.7) | 742 (8.8) |
| > = 55 | 351 (15.6) | 169 (6.8) | 359 (9.7) | 879 (10.4) |
| Unknown | 0 (0.0) | 0 (0.0) | 6 (0.1) | 6 (0.1) |
| TB History Category | | | | |
| New | 1871 (83.1) | 2134 (85.4) | 3277 (88.7) | 7282 (86.3) |
| Re-treatment | 282 (12.5) | 238 (9.5) | 277 (7.5) | 797 (9.4) |
| Transfer in | 99 (4.4) | 128 (5.1) | 140 (3.8) | 367 (4.3) |
| HIV Status of TB Cases | | | | |
| Non-infected | 1627 (72.2) | 1920 (76.8) | 2812 (76.1) | 6359 (75.3) |
| Infected | 532 (23.6) | 476 (19.0) | 715 (19.4) | 1723 (20.4) |
| Unknown | 93 (4.1) | 104 (4.2) | 167 (4.5) | 364 (4.3) |
| Year of Registration | | | | |
| 2015 | 726 (32.2) | 585 (23.4) | 976 (26.4) | 2287 (27.1) |
| 2016 | 668 (29.7) | 536 (21.4) | 946 (25.6) | 2150 (25.5) |
| 2017 | 458 (20.3) | 636 (25.4) | 874 (23.7) | 1968 (23.3) |
| 2018 | 400 (17.8) | 743 (29.7) | 898 (24.3) | 2041 (24.2) |
| Sub-cities in Addis Ababa | | | | |
| Addis Ketema | 147 (6.5) | 144 (5.8) | 236 (6.4) | 527 (6.2) |
| Kolfe Keranyo | 610 (27.1) | 720 (28.8) | 1003 (27.2) | 2333 (27.6) |
| Bole | 122 (5.4) | 148 (5.9) | 205 (5.5) | 475 (5.6) |
| Akaki-Kaliti | 176 (7.8) | 196 (7.8) | 229 (6.2) | 601 (7.1) |
| Yeka | 606 (26.9) | 506 (20.2) | 753 (20.4) | 1865 (22.1) |
| Arada | 27 (1.2) | 37 (1.5) | 83 (2.2) | 147 (1.7) |
| Nefas Silk | 518 (23.0) | 701 (28.0) | 984 (26.6) | 2203 (26.1) |
| Kirkos | 46 (2.0) | 48 (1.9) | 201 (5.4) | 295 (3.5) |
| **Total** | **2252 (26.7)** | **2500 (29.6)** | **3694 (43.7)** | **8446 (100)** |

PTBN—Pulmonary TB bacteriologically negative; PTBP—Pulmonary TB bacteriologically positive; EPTB—Extra-pulmonary TB

## 3.5 Treatment success and associated factors

After adjusting factors associated with treatment outcome, age categories 35–44 (AOR: 0.49, 95% CI: 0.27–0.90), 45–54 (AOR: 0.36, 95% CI: 0.19–0.67), > = 55 (AOR: 0.20, 95% CI: 0.11–0.37) and TB/HIV co-infected patients (AOR: 0.44 95% CI: 0.36–0.53) were predicted with lower odds of successful treatment outcome. Whereas patients receiving treatment at Amoraw Health Center in Bole Sub-city had a better treatment success rate (AOR: 1.73, 95% CI: 1.03–2.91) compared to Addis Ketema Health Center in Addis Ketema Sub-city Table 5.

**Table 4. Treatment outcomes of TB patients at Addis Ababa, Ethiopia between 2015 and 2018.**

| Variables | Treatment Outcome[a] in Number (%) | | | | | Total N (%) | P-value |
|---|---|---|---|---|---|---|---|
| | Successful | | Unsuccessful | | | | |
| | Cured | T. Completed | Died | T. Failed | Loss to Follow-Up | | |
| Sex | | | | | | | |
| F | 965 (24.7) | 2676 (68.6) | 22 (0.6) | 196 (5.0) | 44 (1.1) | 3903 (48.3) | < 0.001 |
| M | 1240 (29.7) | 2597 (62.2) | 26 (0.6) | 222 (5.3) | 92 (2.2) | 4177 (51.7) | |
| Age Category in years | | | | | | | |
| < 15 | 48 (15.5) | 249 (80.6) | 1 (0.3) | 8 (2.6) | 3 (1.0) | 309 (3.8) | < 0.001 |
| 15–24 | 689 (30.9) | 1439 (64.5) | 20 (0.9) | 44 (2.0) | 38 (1.7) | 2230 (27.6) | |
| 25–34 | 789 (30.5) | 1662 (64.3) | 11 (0.4) | 86 (3.3) | 37 (1.4) | 2585 (32.0) | |
| 35–44 | 366 (25.7) | 922 (64.7) | 9 (0.6) | 104 (7.3) | 24 (1.7) | 1425 (17.6) | |
| 45–54 | 175 (24.9) | 444 (63.2) | 2 (0.3) | 66 (9.4) | 15 (2.1) | 702 (8.7) | |
| > = 55 | 138 (16.8) | 552 (67.1) | 5 (0.6) | 110 (13.4) | 18 (2.2) | 823 (10.2) | |
| Unknown | 0 (0.0) | 5 (83.3) | 0 (0.0) | 0 (0.0) | 1 (16.7) | 6 (0.1) | |
| TB History Category | | | | | | | |
| New | 1916 (27.3) | 4604 (65.6) | 45 (0.6) | 344 (4.9) | 109 (1.6) | 7018 (86.9) | 0.005 |
| Re-treatment | 189 (25.9) | 468 (64.0) | 2 (0.3) | 56 (7.7) | 16 (2.2) | 731 (9.0) | |
| Transfer in | 100 (30.2) | 201 (60.7) | 1 (0.3) | 18 (5.4) | 11 (3.3) | 331 (4.1) | |
| TB Type | | | | | | | |
| PTBN | NA | 1994 (91.6) | 7 (0.3) | 140 (6.4) | 35 (1.6) | 2176 (26.9) | < 0.001 |
| PTBP | 2205 (92.9) | 0 (0.0) | 29 (1.2) | 82 (3.5) | 58 (2.4) | 2374 (29.4) | |
| EPTB | NA | 3279 (92.9) | 12 (0.3) | 196 (5.6) | 43 (1.2) | 3530 (43.7) | |
| HIV Status of TB Cases | | | | | | | |
| Non-infected | 1738 (28.4) | 4027 (65.8) | 38 (0.6) | 211 (3.4) | 107 (1.7) | 6121 (75.8) | < 0.001 |
| Infected | 384 (23.5) | 1032 (63.2) | 9 (0.6) | 184 (11.3) | 23 (1.4) | 1632 (20.2) | |
| Unknown | 83 (25.4) | 214 (65.4) | 1 (0.3) | 23 (7.0) | 6 (1.8) | 327 (4.0) | |
| Year of Registration | | | | | | | |
| 2015 | 488 (22.8) | 1452 (67.9) | 17 (0.8) | 134 (6.3) | 49 (2.3) | 2140 (26.5) | <0.001 |
| 2016 | 469 (22.7) | 1453 (70.3) | 11 (0.5) | 100 (4.8) | 35 (1.7) | 2068 (25.6) | |
| 2017 | 575 (30.0) | 1215 (63.4) | 8 (0.4) | 89 (4.6) | 29 (1.5) | 1916 (23.7) | |
| 2018 | 673 (34.4) | 1153 (58.9) | 12 (0.6) | 95 (4.9) | 23 (1.2) | 1956 (24.2) | |
| Sub-cities | | | | | | | |
| Addis Ketema | 119 (23.8) | 337 (67.3) | 5 (1.0) | 29 (5.8) | 11 (2.2) | 501 (6.2) | <0.001 |
| Kolfe Keranyo | 636 (28.2) | 1459 (64.7) | 9 (0.4) | 119 (5.3) | 31 (1.4) | 2254 (27.9) | |
| Bole | 141 (30.5) | 298 (64.4) | 2 (0.4) | 22 (4.8) | 0 (0.0) | 463 (5.7) | |
| Akaki-Kaliti | 175 (29.6) | 369 (62.4) | 8 (1.4) | 29 (4.9) | 10 (1.7) | 591 (7.3) | |
| Yeka | 451 (25.3) | 1195 (66.9) | 11 (0.6) | 99 (5.5) | 30 (1.7) | 1786 (22.1) | |
| Arada | 34 (24.8) | 91 (66.4) | 0 (0.0) | 12 (8.8) | 0 (0.0) | 137 (1.7) | |
| Nefas Silk | 612 (29.1) | 1341 (63.8) | 13 (0.6) | 88 (4.2) | 47 (2.2) | 2101 (26.0) | |
| Kirkos | 37 (15.0) | 183 (74.1) | 0 (0.0) | 20 (8.1) | 7 (2.8) | 247 (3.1) | |
| **Total** | **2205 (27.3)** | **5273 (65.3)** | **48 (0.6)** | **418 (5.2)** | **136 (1.7)** | **8080 (100.0)** | |

[a] TB patient whose treatment outcome is not evaluated/recorded (301) and RR-TB cases transferred to MDR Treatment initiating centers (65) were excluded during analysis. NA- Not applicable; T. completed- treatment completed; T. failed- treatment failed; **PTBN**—Pulmonary TB bacteriologically negative; **PTBP**—Pulmonary TB bacteriologically positive; **EPTB**—Extra-pulmonary TB

**Table 5. Factors associated with successful treatment outcomes of TB patients in Addis Ababa, Ethiopia between 2015 and 2018.**

| Variables | Total N (%) | Treatment success N (%) | COR (95% CI) | AOR (95% CI) | p-value |
|---|---|---|---|---|---|
| Sex | | | | | |
| F | 3903 (48.3) | 3641 (93.3) | 1 | 1 | |
| M | 4177 (51.7) | 3837 (91.9) | **0.81 (0.69–0.96)** | 0.90 (0.76–1.08) | 0.268 |
| Age | | | | | |
| 0–14 | 309 (3.8) | 297 (96.1) | 1 | 1 | |
| 15–24 | 2230 (27.6) | 2128 (95.4) | 0.84 (0.46–1.55) | 0.76 (0.41–1.40) | 0.378 |
| 25–34 | 2585 (32.0) | 2451 (94.8) | 0.74 (0.40–1.35) | 0.77 (0.42–1.42) | 0.407 |
| 35–44 | 1425 (17.6) | 1288 (90.4) | **0.38 (0.21–0.69)** | **0.49 (0.27–0.90)** | **0.021** |
| 45–54 | 702 (8.7) | 619 (88.2) | **0.30 (0.16–0.56)** | **0.36 (0.19–0.67)** | **0.001** |
| > = 55 | 823 (10.2) | 690 (83.8) | **0.21 (0.11–0.38)** | **0.20 (0.11–0.37)** | **<0.001** |
| Unknown | 6 (0.1) | 5 (83.3) | 0.20 (0.02–1.87) | 0.23 (0.02–2.20) | 0.201 |
| TB History Category | | | | | |
| New | 7018 (86.9) | 6520 (92.9) | 1 | 1 | |
| Re-treatment | 731 (9.0) | 657 (89.9) | **0.68 (0.52–0.88)** | 0.90 (0.69–1.18) | 0.451 |
| Transfer in | 331 (4.1) | 301 (90.9) | 0.77 (0.52–1.13) | 0.78 (0.53–1.17) | 0.229 |
| TB Type | | | | | |
| PTBN | 2176 (26.9) | 1994 (91.6) | 1 | 1 | |
| PTBP | 2374 (29.4) | 2205 (92.9) | 1.19 (0.96–1.48) | 0.93 (0.74–1.16) | 0.504 |
| EPTB | 3530 (43.7) | 3279 (92.9) | 1.19 (0.98–1.46) | 0.98 (0.80–1.21) | 0.863 |
| HIV Status of TB Cases | | | | | |
| Non-infected | 6121 (75.8) | 5765 (94.2) | 1 | 1 | |
| Infected | 1632 (20.2) | 1416 (86.8) | **0.41 (0.34–0.48)** | **0.44 (0.36–0.53)** | **<0.001** |
| Unknown | 327 (4.0) | 297 (90.8) | **0.61 (0.41–0.90)** | **0.60 (0.40–0.91)** | **0.016** |
| Year of Registration | | | | | |
| 2015 | 2140 (26.5) | 1940 (90.7) | 1 | 1 | |
| 2016 | 2068 (25.6) | 1922 (92.9) | **1.36 (1.09–1.70)** | **1.39 (1.10–1.75)** | **0.006** |
| 2017 | 1916 (23.7) | 1790 (93.4) | **1.47 (1.16–1.85)** | **1.42 (1.12–1.80)** | **0.004** |
| 2018 | 1956 (24.2) | 1826 (93.4) | **1.45 (1.15–1.82)** | **1.43 (1.13–1.81)** | **0.003** |
| Sub-cities | | | | | |
| Addis Ketema | 501 (6.2) | 456 (91.0) | 1 | 1 | |
| Kolfe Keranyo | 2254 (27.9) | 2095 (92.9) | 1.30 (0.92–1.84) | 1.30 (0.91–1.85) | 0.148 |
| Bole | 463 (5.7) | 439 (94.8) | **1.81 (1.08–3.01)** | **1.73 (1.03–2.91)** | **0.039** |
| Akaki-Kaliti | 591 (7.3) | 544 (92.0) | 1.14 (0.75–1.75) | 1.21 (0.78–1.88) | 0.387 |
| Yeka | 1731 (21.4) | 1595 (92.1) | 1.16 (0.82–1.65) | 1.21 (0.85–1.75) | 0.289 |
| Arada | 192 (2.4) | 176 (91.7) | 1.09 (0.60–1.97) | 1.38 (0.75–2.55) | 0.299 |
| Nefas Silk | 2101 (26.0) | 1953 (93.0) | 1.30 (0.92–1.85) | 1.28 (0.90–1.84) | 0.171 |
| Kirkos | 247 (3.1) | 220 (89.1) | 0.80 (0.49–1.33) | 1.13 (0.67–1.92) | 0.650 |

Bold- values, which show significance (< 0.05); COR- crude odd ratio; AOR- adjusted odd ratio; **PTBN**—Pulmonary TB bacteriologically negative; **PTBP**—Pulmonary TB bacteriologically positive; **EPTB**—Extra-pulmonary TB

## 4. Discussion

Early diagnosis of presumptive TB cases has great importance for improving the treatment outcome and reducing the risk of transmitting the disease to others. Xpert MTB/RIF assay has higher sensitivity than smear microscopy in detecting TB cases [16–21]. This study was to evaluate the TB diagnosis and treatment outcome in health facilities of Addis Ababa, Ethiopia after the introduction of the GeneXpert system.

The introduction of the Xpert MTB/RIF assay in Addis Ababa increased the number of confirmed TB cases by four-fold in 2018 with reference to 2015. This might be due to the high sensitivity of Xpert MTB/RIF assay in *M. tuberculosis* detection [16–21]. This finding is supported by the logistic regression analysis result of the current study. Presumptive TB cases diagnosed with GeneXpert had 4.78 odds of being TB positive compared to those that were diagnosed by microscopy. This finding was consistent with the previous studies performed on the advanced diagnostic performance of the GeneXpert system over smear microscopy that revealed a higher TB case detection [18, 22–27].

Male patients who were screened with TB had higher odds (AOR: 1.45, 95% CI: 1.36–1.54) of being TB positive compared to females. This finding is concordant with findings from the 2016 Global Burden of Diseases and different studies conducted in different countries, particularly in low and middle-income countries [28–34]. Such sex differences might be related to biological TB susceptibility differences between males and females [35], but as indicated by Manish Gupta the underlying genetic mechanism is not clearly defined [36]. 62.8% of patients screened for TB were in the economically productive age group (15–44 years), and they had higher odds of being positive compared to the age group less than 14 years. This might be due to the travel history of the economically productive age group far from their home looking for education, jobs, and social interactions. These might increase the chance of being in contact with infected individuals. The other factor associated with children less than 14 years of age might be the sputum quality, since children are less likely to produce good-quality sputum [37].

In this finding, a 30% drop in death rate was seen in 2018 among patients on anti-TB treatment compared with 2015. The trends in treatment success between consecutive years showed a relative improvement in each year between 2015 and 2018. Unlike findings from reviews conducted to see the impact of Xpert MTB/RIF assay on the treatment outcome from individual patient data meta-analysis [38, 39], in this study an improvement in case detection and success rate of treatment outcomes was observed. The global priority indicator and target for all countries were planned to reach a 90% treatment success rate by the year 2025 at the latest [40]. In this regard, the End TB Strategy towards the TB treatment success rate was met early in 2018 in Addis Ababa, Ethiopia. Of all TB types, 92.6% of TB patients had successful treatment outcomes. Among pulmonary TB-positive cases, 92.9% were cured. The mean treatment success rate in this study was higher than treatment success rates reported in districts of Southwestern (88%), Asela Teaching Hospital (81.7%), and in the refugee camps (75.9%) in Ethiopia [41–43]. But relatively lower than the 96% treatment outcome notified for new TB cases in Ethiopia in 2017 [44].

The male patients enrolled for DOTs service in the study period were 51.7%, which is comparable to reports from selected health centers in Addis Ababa 51% [45]. It is also found lower than findings from different parts of Ethiopia; 53.8% in Asella Teaching Hospital [42], 58% in Mekele [46], while slightly higher than 46.8% in Addis Ababa [47]. Among the total TB cases that received anti-TB treatment 20.4% of TB cases were HIV-positive. This was slightly higher than the HIV-positive rate reported in Northern Ethiopia 18.8% [46] but lower than among notified TB cases in 2016 in Addis Ababa 28% [48].

The odds of treatment outcome in age groups and TB/HIV co-infection in this study was in agreement with findings from Addis Ababa, Asella Teaching Hospital, and Debre Tabor General Hospital [42, 45, 49]. TB patients with ages > = 35 years and being co-infected with HIV were likely to have a lower treatment success rate. This might be due to non-adherence to TB treatment as a result of possible factors such as family responsibility and being busy with socio-economic activities [50]. In TB/HIV co-infected patients, it might be due to the TB and ART medications taken simultaneously which leads to drug-drug interactions, overlapping drug toxicities, and immune reconstitution syndrome [51]. Patients receiving anti-TB treatment at Amoraw HC in

Bole Sub-city were predicted to have higher positive treatment outcomes compared to Addis Ketema HC in Addis Ketema Sub-city. This might be due to patients at Bole Sub-city having relatively good socio-economic and housing status compared to Addis Ketema Sub-city [52, 53].

This study was able to evaluate the significance of diagnosing presumptive TB cases with GeneXpert system over microscopy by including 45,158 presumptive PTB cases and the TB treatment outcome by assessing records of 8080 TB cases in the study area. However, it has the following limitations: it did not include RR-TB cases transferred to MDR Treatment initiating centers and bacteriologically confirmed PTB cases which were diagnosed in two health facilities without TB treatment unit in the treatment outcome analysis; second due to the nature of a retrospective secondary data, there were missing data like sex and age categories. However, as we have included more health facilities and a large number of TB cases in the study, the effect of missing data was considered not to be significant; third this study did not assess other factors or actions taken in health facilities to improve treatment outcome, which might have a positive role in treatment outcomes improvement in addition to GeneXpert system. Lastly, this study lacks comorbidities other than HIV such as diabetes, hypertension, obesity, cancer etc which might have an effect on TB treatment outcome.

## 5. Conclusion

The introduction of the GeneXpert system in Addis Ababa, Ethiopia increased the number of confirmed TB cases with odds of 4.78 compared to smear microscopy. The success of treatment outcomes steadily improved each year in the study period. The overall success rate was 92.6%. Therefore, scaling up the Xpert MTB/RIF assay as a point-of-care test for presumptive TB cases in resource-limited settings would have a significant impact on meeting the SDG and End TB strategy both in TB detection and anti-TB treatment outcome.

## Supporting information

**S1 Data.**
(XLSX)

## Acknowledgments

The authors would like to thank the Institutes of Hygiene and Public Health and of Medical Microbiology, Immunology and Parasitology, University Hospital Bonn and Addis Ababa Public Health Research and Emergency Management Ethical Review Office for reviewing the study protocol and writing a support letter for health facilities. We would like to thank Tewodros Daniel, Ashenafi Alemu, and Alem Mekonnen for their unreserved cooperation throughout the course of data collection. We are also grateful to health professionals working at the study sites because without their willingness, cooperation, and support it would not be possible to come up with these findings.

## Author Contributions

**Conceptualization:** Desalegn Addise Getahun.

**Data curation:** Desalegn Addise Getahun.

**Formal analysis:** Desalegn Addise Getahun, Laura E. Layland, Achim Hoerauf, Biniam Wondale.

**Funding acquisition:** Achim Hoerauf.

**Investigation:** Desalegn Addise Getahun, Laura E. Layland, Achim Hoerauf, Biniam Wondale.

**Methodology:** Desalegn Addise Getahun, Laura E. Layland, Achim Hoerauf, Biniam Wondale.

**Project administration:** Desalegn Addise Getahun.

**Resources:** Desalegn Addise Getahun.

**Software:** Desalegn Addise Getahun, Laura E. Layland, Achim Hoerauf, Biniam Wondale.

**Supervision:** Desalegn Addise Getahun, Laura E. Layland, Achim Hoerauf, Biniam Wondale.

**Validation:** Desalegn Addise Getahun, Laura E. Layland, Achim Hoerauf, Biniam Wondale.

**Writing – original draft:** Desalegn Addise Getahun.

**Writing – review & editing:** Desalegn Addise Getahun, Laura E. Layland, Achim Hoerauf, Biniam Wondale.

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
