## [Decision Letter · Decision Letter 0]

12 Apr 2022

PONE-D-21-40508Impact of GeneXpert on TB Diagnosis and Anti-TB Treatment Outcome at Health Facilities in Addis Ababa, Ethiopia: In the Post MDG YearsPLOS ONE

Dear Dr. Getahun,

Thank you for submitting your manuscript to PLOS ONE. After careful consideration, we feel that it has merit but does not fully meet PLOS ONE’s publication criteria as it currently stands. Therefore, we invite you to submit a revised version of the manuscript that addresses the points raised during the review process.

This manuscript has potential value but there are major issues which should be addressed. There needs to be extensive English language editing including ensuring terminology like the references to the geneXpert are correct. In addition, there was some concern that the treatment outcomes were the incorrect measure of utility.The authors should also defined how they assess sensitivity and specificity performance

We look forward to receiving your revised manuscript.

Kind regards,

Elizabeth S. Mayne, M.D.

Academic Editor

PLOS ONE

Journal Requirements:

2. We note that Figure (1) in your submission contain copyrighted images. All PLOS content is published under the Creative Commons Attribution License (CC BY 4.0), which means that the manuscript, images, and Supporting Information files will be freely available online, and any third party is permitted to access, download, copy, distribute, and use these materials in any way, even commercially, with proper attribution. For more information, see our copyright guidelines: http://journals.plos.org/plosone/s/licenses-and-copyright.

1. You may seek permission from the original copyright holder of Figure (1) to publish the content specifically under the CC BY 4.0 license. 

Reviewers' comments:

Reviewer's Responses to Questions

**Comments to the Author**

1. Is the manuscript technically sound, and do the data support the conclusions?

Reviewer #1: Yes

Reviewer #2: Partly

2. Has the statistical analysis been performed appropriately and rigorously? 

Reviewer #1: Yes

Reviewer #2: I Don't Know

3. Have the authors made all data underlying the findings in their manuscript fully available?

Reviewer #1: Yes

Reviewer #2: Yes

4. Is the manuscript presented in an intelligible fashion and written in standard English?

Reviewer #1: No

Reviewer #2: No

5. Review Comments to the Author

Reviewer #1: Overall comments

1. The manuscript doesn’t read well so I suggest scientific editing. Word usage is incorrect eg. In the abstract, GeneXpert is described as an effective molecular diagnostic test, yet the GeneXpert is an analyzer, not a test.

2. Some sentences are too long and difficult to follow eg. Sentence 2 in the Background section.

Specific comments

1. Background:

a. Second paragraph, Line 4: Not sure how “acceptable” and “excellent” descriptions for the sensitivity and specificity were assigned. Please clarify.

b. Second paragraph, line 8: is the word “TB” supposed to follow the word “paucibacillary?”

2. Study design and data collection

a. Line 8: Five health centres were excluded due to refusal – not sure what this refers to.

3. Demographics and clinical characteristics of study participant on anti-TB treatment

a. First paragraph is a repetition of information provided in the “Study design and data collection” section

b. Was the Xpert MTB/RIF assay used to diagnose pTB as well as EPTB?

c. The conclusion reached is that the number of patients who started TB treatment increased due to introduction of the Xpert MTB/RIF assay but its not clear how many patients were also started on treatment due to other reasons e.g. positive culture or clinical case definitions? Was this assessed?

Reviewer #2: The authors have reviewed retrospective data since the implementation of the Xpert assay for TB diagnosis at participating facilities in Addis Ababa. The aim was to determine the impact of Xpert on TB diagnosis compared to smear microscopy, which is expected as the sensitivity far exceeds that microscopy. Secondly, they further assessed the impact on treatment outcomes since the introduction of the technology - measuring the impact is vague as there are several other factors to be considered for treatment outcome and attributing outcomes to a diagnostic may not be the conclusive. The authors should have rather assessed impact on treatment initiation rather than outcome. The focus of the papers starts shifting between different variables not specifically linked to the Xpert testing. The paper structure needs review to improve the flow of information to provide the supporting evidence showing the impact of Xpert on both TB diagnosis and treatment outcomes.

6. PLOS authors have the option to publish the peer review history of their article (what does this mean?). If published, this will include your full peer review and any attached files.

Reviewer #1: No

Reviewer #2: No

---

## [Author Response · Author response to Decision Letter 0]

15 Jul 2022

Response to Reviewer 1 Comments

1. The manuscript doesn’t read well so I suggest scientific editing. Word usage is incorrect eg. In the abstract, GeneXpert is described as an effective molecular diagnostic test, yet the GeneXpert is an analyzer, not a test. 

Response: Thank you for the comments and we made corrections per the comments.

2. Some sentences are too long and difficult to follow eg. Sentence 2 in the Background section. 

 a. Background: Second paragraph, Line 4: Not sure how “acceptable” and “excellent” descriptions for the sensitivity and specificity 

 were assigned. Please clarify. 

 Response: Thank you for the comments and we made corrections to the paragraph per the comments.

 b. Background: Second paragraph, line 8: is the word “TB” supposed to follow the word “paucibacillary?”

 Response: Thank you for the suggestion. We made a correction per the suggestion.

3. Study design and data collection

Line 8: Five health centres were excluded due to refusal – not sure what this refers to. 

Response: Thank you for the suggestion. We made a correction per the suggestion. To be specific during the data collection period we were challenged by three independent problems those are (1) Unable to get laboratory TB registry log book in four health facilities, (2) the health facility representative of one health facility refused to participate in the study

4. Demographics and clinical characteristics of study participant on anti-TB treatment: First paragraph is a repetition of information provided in the “Study design and data collection” section. 

Response: Thank you for the suggestion. We made a correction per the suggestion.

5. Demographics and clinical characteristics of study participant on anti-TB treatment: Was the Xpert MTB/RIF assay used to diagnose pTB as well as EPTB? 

Response: Thank you for the curiosity. According to the data collected from the TB DOTs register we have seen that two EPTB cases were diagnosed by Xpert MTB/RIF assay, however, we didn’t get extra-pulmonary samples diagnosed by Xpert MTB/RIF assay from the TB laboratory register.

Response to Reviewer 2 Comments

1.The authors have reviewed retrospective data since the implementation of the Xpert assay for TB diagnosis at participating facilities in Addis Ababa. The aim was to determine the impact of Xpert on TB diagnosis compared to smear microscopy, which is expected as the sensitivity far exceeds that microscopy. Secondly, they further assessed the impact on treatment outcomes since the introduction of the technology - measuring the impact is vague as there are several other factors to be considered for treatment outcome and attributing outcomes to a diagnostic may not be the conclusive. The authors should have rather assessed impact on treatment initiation rather than outcome. The focus of the papers starts shifting between different variables not specifically linked to the Xpert testing. 

Response:Thank you for the comments. It is totally acceptable. In this study we have seen that an increased number of TB cases were diagnosed since the introduction of GeneXpert system for the diagnosis of all presumptive TB cases at the participating health facilities in Addis Ababa. There might be several factors that negatively or positively affect the treatment outcome; however, our intention was to see the trends in number of TB cases detected and the positive treatment outcome in the study period compared to the years before the introduction of GeneXpert system. Since as it is clearly said in the comment, there could be several other factors. So, we accept this comment and kept as one of the study limitation in the last paragraph of discussion section. 

2. The paper structure needs review to improve the flow of information to provide the supporting evidence showing the impact of Xpert on both TB diagnosis and treatment outcomes. 

Response: Thank you for the suggestion. We made a correction per the suggestion.

---

## [Editor Report · Decision Letter 1]

10 Jan 2023

PONE-D-21-40508R1Impact of GeneXpert on TB Diagnosis and Anti-TB Treatment Outcome at Health Facilities in Addis Ababa, Ethiopia: In the Post MDG Years

PLOS ONE

Dear Dr. Getahun,

Thank you for submitting your manuscript to PLOS ONE. After careful consideration, we feel that it has merit but does not fully meet PLOS ONE’s publication criteria as it currently stands. Therefore, we invite you to submit a revised version of the manuscript that addresses the points raised during the review process.

A number of improvements have been made in response to the reviewers queries. There are, however, still some significant issues which require clarification.

1. The introduction speaks of the ability of the GeneXpert to detect rifampicin resistance but these data are not included and it is unclear how the presence of RR is correlated with other measures. This is a serious flaw throughout the manuscript.

2. The methodology is incomplete - please discuss the specimens utilised (especially for the 0-14 age range) - were gastric washings, urine, blood and other samples included or just sputa. What was the failure rate of the equipment and what controls and monitoring were run. What was the reflex testing for culture and did any samples test positive on culture but negative on microscopy or geneXpert. What was the clinical correlation in these cases. 

3. Treatment outcome is defined as successful if the patient was cured or the treatment was completed. Please clarify how "cure" was defined in the 27.3% of patients who were cured. In the high number of individuals who completed treatment only, what was the outcome if they were not cured? What is the definition of treatment failure (were they bacteriologically still positive

4. The age range is suggestive of an economically active population. Was there any association with economic activity. The HIV statistics should be clarified in the context of the Ethiopian statistics as well.

5. Please change HIV reactive and unreactive to infected. Please also indicate where known whether patients were virologically suppressed or on ART (if known) and indicate whether this has a relationship to treatment outcome.

6. Please confirm what transfer in indicates

7. Were all treated patients and retreated patients on a standard 6 month regimen? If not, how many were on second line therapy. Were these patients admitted?

8. The following should be included in the discussion: why was there a slight male preponderance, can you hypothesise as to the age spread, can you comment on the efficacy of the test, can you comment on the acuracy of the test, can you link treatment response to HIV status, is there any other comorbid condition (diabetes, hypertension, obesity, cancer etc) which may explain the results. If these data are not available, that represents a significant limitations which should be addressed. In short there is a lack of explanation of the relevant findings which must be addressed.

A minor comment is that the manuscript still requires significant English language editing.

We look forward to receiving your revised manuscript.

Kind regards,

Elizabeth S. Mayne, M.D.

Academic Editor

PLOS ONE

Additional Editor Comments (if provided):

A number of improvements have been made in response to the reviewers queries. There are, however, still some significant issues which require clarification.

1. The introduction speaks of the ability of the GeneXpert to detect rifampicin resistance but these data are not included and it is unclear how the presence of RR is correlated with other measures. This is a serious flaw throughout the manuscript.

2. The methodology is incomplete - please discuss the specimens utilised (especially for the 0-14 age range) - were gastric washings, urine, blood and other samples included or just sputa. What was the failure rate of the equipment and what controls and monitoring were run. What was the reflex testing for culture and did any samples test positive on culture but negative on microscopy or geneXpert. What was the clinical correlation in these cases.

3. Treatment outcome is defined as successful if the patient was cured or the treatment was completed. Please clarify how "cure" was defined in the 27.3% of patients who were cured. In the high number of individuals who completed treatment only, what was the outcome if they were not cured? What is the definition of treatment failure (were they bacteriologically still positive

4. The age range is suggestive of an economically active population. Was there any association with economic activity. The HIV statistics should be clarified in the context of the Ethiopian statistics as well.

5. Please change HIV reactive and unreactive to infected. Please also indicate where known whether patients were virologically suppressed or on ART (if known) and indicate whether this has a relationship to treatment outcome.

6. Please confirm what transfer in indicates

7. Were all treated patients and retreated patients on a standard 6 month regimen? If not, how many were on second line therapy. Were these patients admitted?

8. The following should be included in the discussion: why was there a slight male preponderance, can you hypothesise as to the age spread, can you comment on the efficacy of the test, can you comment on the acuracy of the test, can you link treatment response to HIV status, is there any other comorbid condition (diabetes, hypertension, obesity, cancer etc) which may explain the results. If these data are not available, that represents a significant limitations which should be addressed. In short there is a lack of explanation of the relevant findings which must be addressed.

A minor comment is that the manuscript still requires significant English language editing.
---

## [Author Response · Author response to Decision Letter 1]

24 Feb 2023

Reviewer comment 1. The introduction speaks of the ability of the GeneXpert to detect rifampicin resistance but these data are not included and it is unclear how the presence of RR is correlated with other measures. This is a serious flaw throughout the manuscript.

Response: We, the authors, would like to acknowledge your concern. When we evaluate the treatment outcome of TB patients receiving anti-TB treatment at the selected health facilities, 65 patients have been transferred to MDR treatment sites because they are diagnosed as RR cases. This is the added value of Xpert MTB/RIF Assay that saves the transferred patients by early diagnosis and also prevented the unnecessary wastage of TB medicines. 

Reviewer comment 2. The methodology is incomplete - please discuss the specimens utilised (especially for the 0-14 age range) - were gastric washings, urine, blood and other samples included or just sputa. What was the failure rate of the equipment and what controls and monitoring were run. What was the reflex testing for culture and did any samples test positive on culture but negative on microscopy or GeneXpert. What was the clinical correlation in these cases. 

Response: Thank you for the suggestion. We made a correction per the suggestion. As indicated in page 3 of the manuscript, since 2017, the GeneXpert system is in use in the capital city of Ethiopia to diagnose all presumptive TB cases presented with signs and symptoms for TB. Which means, before 2017, the first diagnostic method in use was AFB Smear Microscopy. Therefore, we aimed only to see the impact of Xpert MTB/RIF assay on the number of TB cases detected in those years and which had a direct impact on the treatment outcome. To be specific in this study we did not intend to perform any tests rather we collected a retrospective data from the selected health facilities of Addis Ababa. The specimen type available on records for all age group is only sputa and also records of TB culture was not found. For better clarification as commented on methodology part under “Study design, data collection and ethical review” page 4 - we added a sentence which describes the sample source was sputa. Line 6 and 7. 

Reviewer comment 3. Treatment outcome is defined as successful if the patient was cured or the treatment was completed. Please clarify how "cure" was defined in the 27.3% of patients who were cured. In the high number of individuals who completed treatment only, what was the outcome if they were not cured? What is the definition of treatment failure (were they bacteriologically still positive)

Response: Thank you for the comment. 

Operational definitions have been kept for these terms in the methodology part page 5. 

Reviewer comment 4. The age range is suggestive of an economically active population. Was there any association with economic activity. The HIV statistics should be clarified in the context of the Ethiopian statistics as well.

Response: Thank you for the comments. Correction has been made on the age range “15-34” as “15-44”. The HIV co-infection status is clarified against studies conducted in different cities/towns in Ethiopia like in Addis Ababa, Asela and Debre Tabor as indicated on page 14 in paragraph 3.

Reviewer comment 5. Please change HIV reactive and unreactive to infected. Please also indicate where known whether patients were virologically suppressed or on ART (if known) and indicate whether this has a relationship to treatment outcome.

Response: Thank you for the comments and we made corrections per the comments on table 3, 4 and 5 as well as in the respective result summary paragraphs. In this study, we have collected retrospective data from TB lab and treatment logbook in which the virological suppression status of HIV-infected patients was not recorded except HIV reactive and non-reactive.

Reviewer comment 6. Please confirm what transfer in indicates

Response: Thank you for the suggestion. 

Transfer-in is a TB patient category who has been transferred from another health facility at any time to initiate or continue the treatment for the convenient of TB patient. Based on the comment, its definition for clarity has been kept under operational definitions in the methodology part, page 5

Reviewer comment 7. Were all treated patients and retreated patients on a standard 6 month regimen? If not, how many were on second line therapy. Were these patients admitted?

Response: Thank you for the comments. 

• All treated and retreated patients included in the treatment outcome analysis were on a standard 6-month regimen.

• 65 patients who are not included in the treatment outcome analysis as indicated under table 4 on page 10 were transferred out to other health facilities for second-line therapy which are not included in this study since their treatment records are not found in the study health facilities. 

• In general, at national level, TB patients for first line TB treatment are not admitted whereas those who take second line TB treatment are admitted. 

Reviewer comment 8. The following should be included in the discussion: why was there a slight male preponderance, can you hypothesise as to the age spread, can you comment on the efficacy of the test, can you comment on the acuracy of the test, can you link treatment response to HIV status, is there any other comorbid condition (diabetes, hypertension, obesity, cancer etc) which may explain the results. If these data are not available, that represents a significant limitations which should be addressed. In short there is a lack of explanation of the relevant findings which must be addressed.

Response: Thank you for the comments. We made corrections per the comments as indicated on page 13 last paragraph. The secondary data that we collected do not enable us to calculate the efficacy and accuracy of the test and it was not our objective initially. In addition, we didn’t get comorbidities other than HIV co-infection in the TB lab and treatment logbooks. So, we kept the absence of other comorbidities data as a limitation as commented on page 15 last paragraphs.

Reviewer comment 9. A minor comment is that the manuscript still requires significant English language editing.

Response: Thank you for the comments and we made corrections per the comments.

---

## [Decision Letter · Decision Letter 2]

4 Apr 2023

PONE-D-21-40508R2Impact of GeneXpert on TB Diagnosis and Anti-TB Treatment Outcome at Health Facilities in Addis Ababa, Ethiopia: In the Post MDG YearsPLOS ONE

Dear Dr. Getahun,

Thank you for submitting your manuscript to PLOS ONE. After careful consideration, we feel that it has merit but does not fully meet PLOS ONE’s publication criteria as it currently stands. Therefore, we invite you to submit a revised version of the manuscript that addresses the points raised during the review process.

This manuscript still requires significant work:

1. Can the author explain the discrepancy between the date of the implementation of GeneXpert (2017) and the data which include 2015 and 2016

2. Can the author confirm the methodology - here a flow diagram may be useful - outlining the sensitivity and specificity of microscopy as well as the exact staining and review process used

3. This manuscript still requires extensive English language editing

We look forward to receiving your revised manuscript.

Kind regards,

Elizabeth S. Mayne, M.D.

Academic Editor

PLOS ONE

Additional Editor Comments:

This manuscript still requires significant work:

1. Can the author explain the discrepancy between the date of the implementation of GeneXpert (2017) and the data which include 2015 and 2016

2. Can the author confirm the methodology - here a flow diagram may be useful - outlining the sensitivity and specificity of microscopy as well as the exact staining and review process used

3. This manuscript still requires extensive English language editing

Reviewers' comments:

Reviewer's Responses to Questions

**Comments to the Author**

1. If the authors have adequately addressed your comments raised in a previous round of review and you feel that this manuscript is now acceptable for publication, you may indicate that here to bypass the “Comments to the Author” section, enter your conflict of interest statement in the “Confidential to Editor” section, and submit your "Accept" recommendation.

Reviewer #3: All comments have been addressed

Reviewer #4: (No Response)

2. Is the manuscript technically sound, and do the data support the conclusions?

Reviewer #3: Partly

Reviewer #4: No

3. Has the statistical analysis been performed appropriately and rigorously? 

Reviewer #3: I Don't Know

Reviewer #4: I Don't Know

4. Have the authors made all data underlying the findings in their manuscript fully available?

Reviewer #3: No

Reviewer #4: Yes

5. Is the manuscript presented in an intelligible fashion and written in standard English?

Reviewer #3: Yes

Reviewer #4: No

6. Review Comments to the Author

Reviewer #3: Data in Table 1 shows an increase in TB testing (Total), I would like to know the reasons for the increase uptake from 4 635 in year 2015 to 17 572 in Year 2018.

Secondly, the method for diagnosing using smear, was it a single smear per participant? if so why do we see an increase in uptake as GeneXpert testing is introduced.

Reviewer #4: - There are serious issues with style and grammar that affect comprehension and require editing

- There is inconsistency between information given in background and results: the background states that GXP was introduced in 2017, yet GXP data is presented from 2015 onwards

- The background lacks information on microscopy and GXP in terms of performance data/limit of detection

- The method section is flawed; there is no information on MTB testing algorithms in the laboratories; no information on which AFB microscopy method was used; no information on treatment regimens for MDR TB; extracted data and analysis thereof is not clearly stated

- The results section doesn't mention proportion of patients on DS or MDR treatment; patients with extra-pulmonary TB are included yet the methods section only mentions sputum samples-- how were these diagnosed and from which samples?; the impact of the GXP introduction on treatment outcomes is not mentioned in results and should be elaborated on; relevance of adding data for various 'sub-cities' not clear unless of local interest

- Comments by previous reviewers have not been adequately addressed

7. PLOS authors have the option to publish the peer review history of their article (what does this mean?). If published, this will include your full peer review and any attached files.

Reviewer #3: No

Reviewer #4: No

---

## [Author Response · Author response to Decision Letter 2]

15 May 2023

1. Can the author explain the discrepancy between the date of the implementation of GeneXpert (2017) and the data which include 2015 and 2016

Response: Thank you for the comments. 

We, the authors, would like to make clear the discrepancy related to the implementation of GeneXpert in Ethiopia;

• It is true that the GeneXpert instruments were introduced in Ethiopia and in use since 2012 (Alemu et al., 2019), to diagnose only selected presumptive TB cases (Presumptive DR-TB, HIV +ve, Children; and EPTB), while the remaining cases with low-risk to DR-TB, HIV-VE, and adult were diagnosed by AFB microscopy based on the algorithm (FMoH Ethiopia, 2014). But, since 2017, GeneXpert system is in use to diagnose all presumptive TB cases presented with signs and symptoms of TB at any given health facility in Addis Ababa.

• It is because of selective TB diagnosis made by GeneXpert instruments were available in the years 2015 and 2016.

2. Can the author confirm the methodology - here a flow diagram may be useful - outlining the sensitivity and specificity of microscopy as well as the exact staining and review process used

Response: Thank you very much for the comments.

The sensitivity and specificity were not assessed since, in this investigation, we collected retrospective data from health facilities in Addis Ababa.

Both staining techniques (Ziel-Neelsen staining (ZN) or fluorescent auramine staining (LED FM)) were in use in Addis Ababa but the health facilities do not keep track of which staining techniques were employed; instead, the record indicates that the diagnosis was performed using sputum smear microscopy. Due to the lack of clarity in the registration books, we are unable to determine the exact staining techniques.

3. This manuscript still requires extensive English language editing

Response: Thank you for the comments and suggestions, and we made corrections per the comments.

4. Reviewer #3: Data in Table 1 shows an increase in TB testing (Total), I would like to know the reasons for the increase uptake from 4 635 in year 2015 to 17 572 in Year 2018. Secondly, the method for diagnosing using smear, was it a single smear per participant? if so why do we see an increase in uptake as GeneXpert testing is introduced.

Response: Thank you for the comments.

• It is true that since the GeneXpert instrument has been used for diagnosis of presumptive TB cases as a first-line test in the chosen health facilities in Addis Ababa, the number of presumptive TB cases diagnosed for TB has increased. The instrument's improved sensitivity and specificity, which increase the trust of clinicians working in the OPD, ease of use, the lack of a requirement for two sputum samples, and the suitability of the reagent/cartridge may all be contributing factors to the increased TB test.

• Based on the recommendation of the guideline for clinical and programmatic management of TB and TB/HIV in Ethiopia, two sputum samples (Spot-Spot) are required for AFB Microscopy, and one sputum smear positive result confirms TB diagnosis. In this study, we collected the final AFB microscopy result of the patients from the laboratory registration book.

5. Reviewer #4: - There are serious issues with style and grammar that affect comprehension and require editing

- There is inconsistency between information given in background and results: the background states that GXP was introduced in 2017, yet GXP data is presented from 2015 onwards

- The background lacks information on microscopy and GXP in terms of performance data/limit of detection

- The method section is flawed; there is no information on MTB testing algorithms in the laboratories; no information on which AFB microscopy method was used; no information on treatment regimens for MDR TB; extracted data and analysis thereof is not clearly stated

- The results section doesn't mention proportion of patients on DS or MDR treatment; patients with extra-pulmonary TB are included yet the methods section only mentions sputum samples-- how were these diagnosed and from which samples?; the impact of the GXP introduction on treatment outcomes is not mentioned in results and should be elaborated on; relevance of adding data for various 'sub-cities' not clear unless of local interest

Response:Thank you for the comments.

• Thank you for the comments and we made corrections per the comments.

• It is true that the GeneXpert instruments were introduced in Ethiopia and in use since 2012 (Alemu et al., 2019), to diagnose only selected presumptive TB cases (Presumptive DR-TB, HIV +ve, Children; and EPTB), while the remaining cases with low-risk to DR-TB, HIV-VE, and Adult are diagnosed by AFB microscopy based on the algorithm (FMoH Ethiopia, 2014). But, since 2017, GeneXpert system is in use to diagnose all presumptive TB cases presented with signs and symptoms of TB at any given health facility in Addis Ababa. It is because of selective TB diagnoses made by GeneXpert instruments were available in the years 2015 and 2016.

• The performance data is included on page 3 in the last paragraph (we added a sentence which describes the performance of AFB smear microscopy)

• Both staining techniques (Ziel-Neelsen staining (ZN) or fluorescent auramine staining (LED FM)) are in use in Addis Ababa, but the health facilities do not keep track of which staining techniques were employed; instead, the record indicates that the diagnosis was performed using sputum smear microscopy. Due to the lack of clarity in the laboratory registration books, we are unable to determine the exact staining techniques.

• 65 MDR patients who were moved to the MDR Treatment Initiating Center (TIC) for second-line therapy and are not included in this study, as noted under Table 4 on page 10. Therefore, we were unable to obtain their treatment records. As a result, MDR patients are excluded from the treatment outcomes analysis.

• Clinical diagnosis was the main method of diagnosis for extra-pulmonary TB patients. Based on the recorded data only 10 (0.3%) of the 3694 extra-pulmonary TB patients had laboratory diagnostic information, The data record simply states that the diagnosis was made using either smear microscopy or GeneXpert, the health facilities did not keep track of the sample types utilized. That is why we don’t give due attention on the diagnosis of EPTB and we are unable to pinpoint the precise sample type utilized.

• The result section on page 10 and lines 10th and 11th discuss the impact of the GXP introduction on treatment results. 

• We appreciate your concern and your suggestion, adding data for various 'sub-cities' will be pertinent to local interest.

---

## [Decision Letter · Decision Letter 3]

19 Jun 2023

PONE-D-21-40508R3Impact of GeneXpert on TB Diagnosis and Anti-TB Treatment Outcome at Health Facilities in Addis Ababa, Ethiopia: In the Post MDG YearsPLOS ONE

Dear Dr. Getahun,

Thank you for submitting your manuscript to PLOS ONE. After careful consideration, we feel that it has merit but does not fully meet PLOS ONE’s publication criteria as it currently stands. Therefore, we invite you to submit a revised version of the manuscript that addresses the points raised during the review process.

All 3 reviewers suggest that there have been substantial improvments to this manuscript although there are a number of grammatical and typographical errors. As suggested, please also include a more focused description as to the link between the introduction of the GeneXpert and the improvement of TB outcomes. Also, please explain why a number of patients included had "clinical" diagnoses and how these were improved by introduction of the Gene Xpert.==============================

We look forward to receiving your revised manuscript.

Kind regards,

Elizabeth S. Mayne, M.D.

Academic Editor

PLOS ONE

Journal Requirements:

Reviewers' comments:

Reviewer's Responses to Questions

**Comments to the Author**

1. If the authors have adequately addressed your comments raised in a previous round of review and you feel that this manuscript is now acceptable for publication, you may indicate that here to bypass the “Comments to the Author” section, enter your conflict of interest statement in the “Confidential to Editor” section, and submit your "Accept" recommendation.

Reviewer #3: All comments have been addressed

Reviewer #4: (No Response)

Reviewer #5: All comments have been addressed

2. Is the manuscript technically sound, and do the data support the conclusions?

Reviewer #3: Yes

Reviewer #4: Partly

Reviewer #5: Partly

3. Has the statistical analysis been performed appropriately and rigorously? 

Reviewer #3: Yes

Reviewer #4: I Don't Know

Reviewer #5: Yes

4. Have the authors made all data underlying the findings in their manuscript fully available?

Reviewer #3: Yes

Reviewer #4: Yes

Reviewer #5: Yes

5. Is the manuscript presented in an intelligible fashion and written in standard English?

Reviewer #3: Yes

Reviewer #4: No

Reviewer #5: Yes

6. Review Comments to the Author

Reviewer #3: The manuscript has done a thorough analysis of the data and highlighted the important key points in question.

Reviewer #4: Many thanks for addressing some of the previous queries regarding the manuscript.

I would suggest additional work on the following issues

- Style and grammar require further review, there are several phrases and words that are not idiomatic or incorrect. Please also review the tense in which the various sections are written. Some examples highlighted in yellow in attachment.

- Not all abbreviations are explained in the text (see PTBN and PTBP under ‘demographics and clinical characteristics’).

- Should the sentence in the discussion in the third paragraph ‘Male patients who were diagnosed with TB had higher odds of being positive..’ say ‘screened’ rather than ‘diagnosed’? A positive test would lead to the diagnosis, the sentence as it stands doesn’t seem to make sense. This appears again on another occasion in the discussion.

- The discussion still lacks a clear structure and a logical flow, and should focus more on the main aims of the study which were to show a link between increase in Xpert testing and TB diagnosis and treatment outcomes.

The link between introduction of Xpert and improved treatment outcomes is tenous and does not appear to be supported by the data: the treatment outcome group that was analysed appears to include patients that were diagnosed clinically and a large number of patients with extra-pulmonary TB who were also not diagnosed with Xpert it seems (or ?unknown how they are diagnosed). Yet the discussion states that Xpert ‘brought an improvement in…success rate of treatment outcomes’. Please could you clarify how you came to this conclusion? There are recent review articles that have not shown a strong link, ie F Haraka 2021 ‘Impact of the diagnostic test Xpert MTB/RIF on patient outcomes for tuberculosis’ and G Di Tanna 2019 ‘Effect of Xpert MTB/RIF on clinical outcomes in routine care setting: individual patient data meta-analysis’. You do mention in your limitations that other factors that may have had an impact on treatment outcomes were not taken into consideration; I feel this has to be expanded on as it is very relevant. It is mentioned that patients with drug resistant TB were excluded from the analysis so any advantage of picking up rifampicin resistance using the Xpert cartridge on treatment outcomes will not be assessed if I am correct?

Reviewer #5: I think that this is an important subject and provides valuable information on implementation of Gene Xpert, however, there are still a number of errors that need to be addressed for example in the second paragraph of the discussion, they state that there has been a 4 fold increase in identified TB cases, I think the correct term is, confirmed TB cases have increased 4 fold. Similarly, in paragraph 5 of the discussion they state that the cure rate pulmonary TB was 92.9%, they previously should either clarify that they are referring to the PTBP or cite the actual pulmonary TB cure rate which was 27.3%. While the authors have addressed the comments raised by the reviewers there are still some errors that need to be addressed.

7. PLOS authors have the option to publish the peer review history of their article (what does this mean?). If published, this will include your full peer review and any attached files.

Reviewer #3: No

Reviewer #4: No

Reviewer #5: No

---

## [Author Response · Author response to Decision Letter 3]

27 Jun 2023

Reviewer #4: - I would suggest additional work on the following issues

1• Style and grammar require further review, there are several phrases and words that are not idiomatic or incorrect. Please also review the tense in which the various sections are written. Some examples highlighted in yellow in attachment.

Response: Thank you for the comments and suggestions, and we made corrections per the comments and suggestions forwarded

2• Not all abbreviations are explained in the text (see PTBN and PTBP under ‘demographics and clinical characteristics’).

Response: Thank you for the suggestions, and we made corrections per suggestions forwarded

3• Should the sentence in the discussion in the third paragraph ‘Male patients who were diagnosed with TB had higher odds of being positive..’ say ‘screened’ rather than ‘diagnosed’? A positive test would lead to the diagnosis, the sentence as it stands doesn’t seem to make sense. This appears again on another occasion in the discussion.

Response: Thank you for the comments and suggestions, and we made amendment as per the suggestion

4• The discussion still lacks a clear structure and a logical flow, and should focus more on the main aims of the study which were to show a link between increase in Xpert testing and TB diagnosis and treatment outcomes. The link between introduction of Xpert and improved treatment outcomes is tenous and does not appear to be supported by the data: the treatment outcome group that was analysed appears to include patients that were diagnosed clinically and a large number of patients with extra-pulmonary TB who were also not diagnosed with Xpert it seems (or ?unknown how they are diagnosed). Yet the discussion states that Xpert ‘brought an improvement in…success rate of treatment outcomes’. Please could you clarify how you came to this conclusion? There are recent review articles that have not shown a strong link, ie F Haraka 2021 ‘Impact of the diagnostic test Xpert MTB/RIF on patient outcomes for tuberculosis’ and G Di Tanna 2019 ‘Effect of Xpert MTB/RIF on clinical outcomes in routine care setting: individual patient data meta-analysis’. 

Response: • Thank you for the comments and suggestions, and we made corrections per the suggestion. Regarding the conclusions made on the impact of Xpert MTB/RIF Assay on the treatment success rate;

o It is true that this study has limitation related assessing some factors or actions taken in health facilities which might have impact for the improved treatment outcome; however, we have seen that the number of presumptive TB cases screened and diagnosed since the introduction of the Xpert MTB/RIF Assay were significantly increased and also an increased treatment success rate and 30% drop in death rate proves that the introduction of Xpert MTB/RIF Assay as a first diagnostic method for presumptive 

o Additionally, among bacteriologically confirmed pulmonary TB cases, the percent of cured patients were increased to 34.4% in 2018 compared to 22.8% in 2015, since the cure treatment outcome evaluation requires confirmation by diagnostic tests Xpert MTB/RIF assay.

5• You do mention in your limitations that other factors that may have had an impact on treatment outcomes were not taken into consideration; I feel this has to be expanded on as it is very relevant. It is mentioned that patients with drug resistant TB were excluded from the analysis so any advantage of picking up rifampicin resistance using the Xpert cartridge on treatment outcomes will not be assessed if I am correct?

Response: Thank you for the suggestions, and we made corrections per the suggestion that is exclusion of patients with drug resistant TB is included in the limitation section

Reviewer #5: I think that this is an important subject and provides valuable information on implementation of Gene Xpert, however, there are still a number of errors that need to be addressed for example 

Response: Thank you for the comments and suggestions

1• In the second paragraph of the discussion, they state that there has been a 4 fold increase in identified TB cases, I think the correct term is, confirmed TB cases have increased 4 fold. 

Response: Thank you for the comments and suggestions. We made corrections per the comments.

2• Similarly, in paragraph 5 of the discussion they state that the cure rate pulmonary TB was 92.9%, they previously should either clarify that they are referring to the PTBP or cite the actual pulmonary TB cure rate which was 27.3%. While the authors have addressed the comments raised by the reviewers there are still some errors that need to be addressed.

Response: • Thank you for the comments and suggestions. The variation in result among the cured PTBP patient observed is due to the denominator used that means; 

o We used the entire number of patients enrolled as a denominator for the result narrative section on page 9 under ‘demographics and clinical characteristics,’ and for the result table section on table 3, we used the total number of PTBP patients as a denominator.

---

## [Decision Letter · Decision Letter 4]

31 Jul 2023

Impact of the use of GeneXpert on TB diagnosis and anti-TB treatment outcome at health facilities in Addis Ababa, Ethiopia in the post-millennium development years

PONE-D-21-40508R4

Dear Dr. Getahun,

We’re pleased to inform you that your manuscript has been judged scientifically suitable for publication and will be formally accepted for publication once it meets all outstanding technical requirements.

Kind regards,

Elizabeth S. Mayne, M.D.

Academic Editor

PLOS ONE

Additional Editor Comments (optional):

Reviewers' comments:

Reviewer's Responses to Questions

**Comments to the Author**

1. If the authors have adequately addressed your comments raised in a previous round of review and you feel that this manuscript is now acceptable for publication, you may indicate that here to bypass the “Comments to the Author” section, enter your conflict of interest statement in the “Confidential to Editor” section, and submit your "Accept" recommendation.

Reviewer #4: (No Response)

Reviewer #5: All comments have been addressed

2. Is the manuscript technically sound, and do the data support the conclusions?

Reviewer #4: Partly

Reviewer #5: Yes

3. Has the statistical analysis been performed appropriately and rigorously? 

Reviewer #4: I Don't Know

Reviewer #5: Yes

4. Have the authors made all data underlying the findings in their manuscript fully available?

Reviewer #4: Yes

Reviewer #5: Yes

5. Is the manuscript presented in an intelligible fashion and written in standard English?

Reviewer #4: Yes

Reviewer #5: Yes

6. Review Comments to the Author

Reviewer #4: Many thanks for the revisions,

I have highlighted in yellow remaining editing that is suggested.

Additional comments:

- References 6-9 in the background do not support the statement that Xpert improves anti-TB treatment outcomes:

No 6 covers test performance criteria, turn-around-time, time to starting treatment

No 7 is a cost-benefit analysis looking at a cohort in a high income/low burden setting

No 8 is a mathematical model

No 9 addresses practical issues with implementation and time to diagnosis

- Please review paragraph in background addressing previous and current use of Xpert in your area and the tense used, how you were testing previously and how you are testing now; it requires clarity

- Operational definitions: add ‘retreatment’ as mentioned in tables

- Elaborate on 3rd point in discussion: other factors and actions to improve outcome

I still have concerns about causality versus observation with regards to Xpert introduction and treatment outcomes. You have described an increase in use of Xpert and observed an improvement in treatment outcomes over this time period which is important to describe. The diagnosis with Xpert is more accurate and sensitive and should enable patients to be diagnosed and started on an efficient regimen earlier. But in my opinion there are many other factors involved in successful treatment outcomes and further research is required to prove causality.

Reviewer #5: The Authors have made adequate changes to the manuscript and based on that I accept the manuscript as is.

7. PLOS authors have the option to publish the peer review history of their article (what does this mean?). If published, this will include your full peer review and any attached files.

Reviewer #4: No

Reviewer #5: No

---

## [Editor Report · Acceptance letter]

18 Aug 2023

PONE-D-21-40508R4 

Impact of the use of GeneXpert on TB diagnosis and anti-TB treatment outcome at health facilities in Addis Ababa, Ethiopia in the post-millennium development years 

Dear Dr. Getahun:

I'm pleased to inform you that your manuscript has been deemed suitable for publication in PLOS ONE. Congratulations! Your manuscript is now with our production department. 

Kind regards, 

on behalf of

Dr. Elizabeth S. Mayne 

Academic Editor

PLOS ONE